# Selected Properties of Single and Multi-Layered Particleboards with the Structure Modified by Fibers Implication

**DOI:** 10.3390/ma15238530

**Published:** 2022-11-30

**Authors:** Anita Wronka, Piotr Beer, Grzegorz Kowaluk

**Affiliations:** Institute of Wood Science and Furniture, Warsaw University of Life Sciences—SGGW, Nowoursynowska St. 159, 02-776 Warsaw, Poland

**Keywords:** particleboard, fiberboard, particle, fiber, bending, surface roughness, layer, density profile

## Abstract

One of the ways of potential improvement of the particleboard properties, especially surface quality, can be the incorporation of wood fibers to face layers. This study aimed to evaluate the selected mechanical and physical parameters of single and multi-layered particleboards with the structure modified by the incorporation of various types and amounts of wood fibers. Single, 3- and 5-layers particleboards were produced with two different types of wood fibers added to the face and core layers. The basic mechanical parameters (modulus of rupture, modulus of elasticity, internal bond, surface soundness), as well as density profile and surface roughness, have been investigated. The results have shown that the single-layer panels with fibers did not meet the standard requirements due to unsatisfactory unstable parameters, probably caused by uneven resination. The remaining panels, 3- and 5-layer, met the standard requirements, and, due to fiber incorporation, there is also potential to reduce the panel density, still meeting standard requirements. The addition of fibers from 0% to 75% in face layers leads to an increase in the modulus of rupture from 10.6 N mm^−2^ to 15.6 N mm^−2^. Depending on the fibers’ type, the surface soundness can vary between 0.7 N mm^−2^ and 1.2 N mm^−2^. Five-layer panels were of similar or even higher parameters, but due to much-complicated technology, it seems unreasonable to develop this type of composite. The novelty of the conducted research is the attempt to modify the structure of particleboards by adding various amounts of two different types of fibers by mixing them with particles or adding them as separate layers and producing panels of different densities.

## 1. Introduction

The use of thin overlays, like various laminates, melamine-impregnated papers, etc., to cover and finish the particleboards’ surface, forces to keep the surface finished, with high parameters, especially roughness. The quality of the final processed wood-based panel can also be influenced by machining parameters [1], as well as by surface structure. One of the techniques to get a smoother surface is to use fine wood particles to produce face layers of particleboards. In addition, the surface smoothness of wood-based composites can be created by the application of different sizes of wood fibers. However, it should be pointed out that the size of the fibers, measured by the degree of defibration, can influence the mechanical and physical properties of the composites [2]. The higher degree of defibration makes the fibers more useful for insulating boards, where a high amount of small air gaps is profitable to achieve good insulation parameters (since air is a good insulation factor in that kind of material). The lower degree of defibration provides larger fibers, which are necessary to reach higher mechanical strength. A real disadvantage of the application of fine wood particles/fibers is a higher glue demand, which grows when bonded particle size decreases due to an increased specific surface of particles. The profits from good surface quality are available in the special variation of that approach, where the amount of binder is big enough to act as a matrix for the lignocellulosic particles [3]. To improve the surface quality of particleboards, the bagasse fibrous material addition has been implemented [4]. This attempt improved both the mechanical and surface roughness properties of the tested panels. The researchers concluded that, in general, bagasse boards exhibited better surface roughness, due to lower Ra (arithmetic average of the absolute values of the profile heights over the evaluated length) and Rz (average value of the absolute values of the heights of the five highest-profile peaks and the depths of five deepest alleys within the evaluated length) values, than the remaining samples, made with poplar and mixed hardwood particles. It should be pointed out here that when producing the particleboard face layer out of fibrous particles, such a combined material can be evaluated as a medium-density fiberboard regarding surface properties. The reinforcing role of aligned flax and hemp fiber mats applied in the face layers zone of particleboards has been tested by [5]. Depending on the materials used (wood–flax, wood–hemp, hurd–hemp, and shive–flax), the increase of modulus of elasticity during bending was at least 28%. However, weak interfacial bond strength within the flax and hemp fiber layers has been found. The positive influence of the addition of fibrous material (sugarcane bagasse fibers) to face layers of particleboards made of coconut fibers in the core layer has been confirmed by [6]. The opposite effect (linear decrease in bending strength regarding the addition of coir fibers) has been found by [7] when producing rice straw single-layer particleboards with different content (0–100%) of coir fibers. According to Ozdemir et al. [8], one of the key factors influencing the surface properties of the fibrous wood-based composite, like medium-density fiberboard, is the moisture content of the panel, which is strongly connected to ambient environment temperature and humidity. Their findings have shown that the surface adhesion strength of the samples was adversely influenced by high relative humidity exposure. Thus, the surface soundness of tested MDF panels has been reduced from 2.20 N mm^−2^ to 1.10 N mm^−2^, with the panels’ moisture content, increasing from 8.6% to 14%. In addition, the surface roughness, characterized by Ra, has risen from 2.04 μm to 4.91 μm under the same moisture content change. The importance of the moisture influence of the panels covered by fibrous materials has also been confirmed by Ulker [9].

Regarding the surface roughness of fibrous panels for the furniture industry (MDF), not only the surface structure can influence the surface roughness. The results [10] show that using the fine stylus technique, the differences between the roughness parameters can be found when measuring these along and across the sanding direction. It has been found that for some panels taken from the market, these differences reach about 100%.

The surface properties of fibrous wood-based composites can be influenced not only by the use parameters but can be created during panel production. The results of the roughness measurement of MDF produced from fibers impregnated by selected boron-based chemicals have shown that the roughness values of the samples increased with increasing chemical concentration used during panel manufacturing [11]. The use of silane coupling agents to improve the bonding quality between wheat straw particles and urea-formaldehyde (UF) resin also gave a significant improvement in the surface roughness of produced particleboards [12]. An example of a post-production technique of modification of an MDF surface to protect MDF panels against physical and mechanical damage commonly present in real-life indoor conditions is trimethylchlorosilane-SiO_2_ nanoparticles deposition on MDF panels [13]. The particleboards’ structure modification by fibers can lead not only to a surface quality improvement but also to selected mechanical and physical properties modification. According to Hartono et al. [14], the addition of wood shavings improves the elephant dung particleboard’s physical and mechanical properties, especially when the ratio of elephant dung and wood shavings reaches 1:1. The only weakness of the tested panels was moisture content and water absorption, since they did not improve when the panels’ structure has been modified [15]. The panel surface hardness, created by both used material and production parameters, can significantly influence the mechanical processing quality [16].

Summarizing the above-mentioned state-of-art, it can be concluded that in the case of particleboards, several different attempts have been made to change the surface and structure properties, as well as to improve the mechanical and/or physical parameters of the panels by addition of lignocellulosic fibrous materials. However, there is no clear and universal data about the results of that approach. What is more, sometimes the results show an opposite influence of the fibers’ addition to particleboards (for example, improvement or reduction of mechanical parameters of the panels). There is also a lack of information about the influence of the addition of different types of fibers to particleboard structure on the properties of particleboards modified that way. The fibers’ morphological characteristic (degree of defibration/freeness) is crucial in the case of fiberboards and MDF [17,18,19,20]. The incorporation of the fibrous material into particleboards can be completed by adding these by mixing with particles or by separate layers. The potential differences of such approaches are, for example, different density profiles of produced composites, which significantly influence the mechanical properties of the panels, and also various surface roughness parameters, which should be better when creating face layers made of fibers only. However, it should be mentioned that mixing fibers with particles is easier to be done in industrial conditions since no special forming units are needed to form particles and fibers separately. However, mixing fibers with particles during the resination process leads to uncontrolled binder uptake by significantly different materials, as particles and fibers are. In addition, despite the fact that the influence of the density of particleboards and their face layers’ content on their mechanical and physical properties is quite known in the literature [3,4], there is no information available on how these parameters contribute to properties of the particleboards produced with the use of particles and fibers.

To fill this gap, the aim of this study was to evaluate the selected mechanical and physical parameters of single and multi-layered particleboards with the structure modified by the incorporation of wood fibers. In the scope of the research, the influence of type and amount of fibers, way of fibers’ implication (separate layer or mixed with particles), panels’ density, as well as face layers’ share on the panels’ properties has been investigated.

## 2. Materials and Methods

### 2.1. Materials

The industrial particles, pine *Pinus sylvestris* L., about 4% of moisture content (MC), intended to face and core layers particleboard production, received from a plant located in Poland, have been used. Two types of fibers have been tested in investigations: fiber type 1 was taken from the medium-density fiberboard (MDF) industry (IKEA Industry Poland sp. z o.o., Orla, Poland), and fiber type 2 (Research and Development Centre for Wood-Based Panels Sp. z o.o., Czarna Woda, Poland) was taken from fibrous insulating materials. Both plants were located in Poland, and both types of fibers were produced mostly from *Pinus sylvestris* L. The degree of defibration of the fibers, established by producers according to [21] on vibratory sieve shaker Analysette 3 Pro (FRITSCH GmbH Milling and Sizing, Idar-Oberstein, Germany), was as follows: 47 DS for fibers type 1 and 63 DS for fibers type 2. The MC of the fibers was about 3%. The morphological characteristics of the fibers have been presented in Figure 1. As a binder, an industrial urea-formaldehyde (UF) resin Silekol S-123 (Silekol Sp. z o.o., Kędzierzyn—Koźle, Poland) of about 65% of dry content, pH 9.6, the viscosity of 470 mPa s, was used, with 2% of dry matter of ammonium nitrate water solution as a hardener, to reach the curing time of gluing mass in 100 °C about 85 s. The produced boards were conditioned before the tests at 20 °C and 65% air humidity until a constant mass had been obtained.

### 2.2. Elaboration of Panels

The panels of 750 mm × 550 mm, 16 mm nominal thickness, were produced with the resination as follows: 9% for particles and 15% for fibers. The core layer particles and face layers fine particles have been blended in a laboratory glue blender with glue mass separately. In the case of mixed fine particles and fibers for the face layers, these materials have been blended with glue mass, where the resination was tuned accordingly to the mass share of the fibers. The pressing was conducted with a hot press (ZUP-NYSA PH-1P125) at a pressing temperature of 200 °C, press closing time of 10 s, specific maximum unit pressure of 2.5 MPa, and a pressing factor of 10 s mm^−1^ of the nominal thickness of the panel. No calibration (by sanding) has been conducted after pressing the panels. The pictures of the morphology of the surface of the tested composites are shown in Figure 2.

The following constant and variable/changed factors have been investigated (Table 1):-Constant factors: number of layers 1, nominal density 660 kg m^−3^, fibers type 1; variable factor: mass share of fibers 0%, 25%, 50%, and 75% (panel type: 1–4, respectively, to rising fibers share);-Constant factors: number of layers 3, nominal density 660 kg m^−3^, fibers type 1, face layers mass share 16%; variable factor: mass share of fibers in face layers 0%, 25%, 50%, and 75% (panel type: 5–8, respectively, to rising fibers share);-Constant factors: number of layers 3, nominal density 660 kg m^−3^, fibers type 1, face layers mass share 32%; variable factor: mass share of fibers in face layers 0%, 25%, 50%, and 75% (panel type: 9–12, respectively, to rising fibers share);-Constant factors: number of layers 3, fibers type 1, face layers mass share 32%, the mass share of fibers in face layers 50%; variable factor: nominal density 580, 620, 660, and 710 kg m^−3^ (panel type: 13, 14, 11, and 15, respectively, to rising density);-Constant factors: number of layers 5 (large particles in the core, fine particles in the middle layer, fibers on face layers, nominal density 660 kg m^−3^, face layers mass share 32%, the mass share of fibers in face layers 25%; variable factor: fibers type 1 and 2 (panel type: 16 and 17, respectively, to fibers type);-Constant factors: number of layers 3, fibers type 2, face layers mass share 32%, the mass share of fibers in face layers 50%; variable factor: nominal density 580, 620, 660, and 710 kg m^−3^ (panel type: 18, 19, 20, and 21, respectively, to rising density).

### 2.3. Panels’ Examination

The test specimens were cut accordingly to EN 326-2 [22] and EN 326-1 [23]. The modulus of rupture (MOR) and elasticity (MOE) was determined according to EN 310 [24], the internal bond (IB) was determined according to EN 319 [25], and the surface soundness (SS) was measured according to EN 311 [26]. All the mechanical properties were examined with an INSTRON 3369 (Instron, Norwood, MA, USA) laboratory-testing machine, and whenever applicable, the results were referred to standards [27]. The density profiles of the tested PB were measured on a GreCon DAX 5000 device (Fagus-GreCon Greten GmbH & Co. KG, Alfeld/Hannover, Germany). The test of surface roughness (maximum roughness of peaks—Rpm and valleys—Rvm) was performed with Surtronic 25 equipment (Taylor Hobson, Leicester, UK), and the results were calculated as an average value from 10 measurements. For each board type, as many as 10 samples were analyzed. The produced boards were conditioned before the tests in a climatic chamber (producer: Research and Development Centre for Wood-Based Panels Sp. z o.o. in Czarna Woda, Poland) at 20 °C and 65% air humidity until a constant mass was obtained. The achieved results have been collected in Table 2.

### 2.4. Statistical Analysis

Analysis of variance (ANOVA) and *t*-tests calculations were used to test (α = 0.05) for significant differences between factors and levels, where appropriate, using IBM SPSS statistic base (IBM, SPSS 20, Armonk, NY, USA). A comparison of the means was performed when the ANOVA indicated a significant difference by employing the Duncan test. The statistically significant differences in achieved results are given in the Results and Discussion paragraph whenever the data were evaluated.

## 3. Results and Discussion

### 3.1. Modulus of Rupture and Modulus of Elasticity

The results of the modulus of rupture investigation have been displayed in Figure 3. As it can be seen, in the case of rising fiber share in the single-layer panels (panel type 1–4), there is no remarkable relation between MOR and fiber content. Additionally, high standard deviation values (error bars on the plot) indicate high dispersion of individual samples MOR. This can be caused by the irregular location of fibers and particles, along with the thickness, as well as along the wide surfaces (Figure 2(1–4)). The lowest MOR value, 5.30 N mm^−2^, statistically significantly different from panels 1, 2, and 4, found for panel 3, can be influenced by the low density of the face layers since these are responsible for carrying the strains during bending. The same explanation applies to panel 2, where the highest MOR (14.5 N mm^−2^) has been reached. This phenomenon has been confirmed by [23]. When testing three-layer panels with 16% of face layers, a strong correlation between rising fiber content and increasing MOR (from 10.6 N mm^−2^ to 15.6 N mm^−2^) has been found (panel type 5–8). In these panels, comparably low content of face layers (16%) contains rising content of fibers, which, retained in outer zones, significantly improve bending properties. This phenomenon is strongly connected to the density of the face layers of panels 5–8, where the increase in density of these zones is well visible. Among this group, there are statistically significant differences between MOR values of panels 5 and 6–8, as well as between panels 6 and 8. When increasing the face layers’ share to 32% (panels 9–12), the rise of the fiber content in face layers causes a rise in MOR values. However, a statistically significant increase can be found when fiber content rises from 0% (panel 9) to 25% (panel 10). The remaining panels’ MOR values remain on a similar level. Such stabilization of MOR values for panels 10–12 is due to the similar density profiles of the face layers of these panels. When investigating the influence of the panel’s density with constant 50% fiber content (panels 13, 14, 11, 15), it can be concluded that the density increase in the range of 580–710 kg m^−3^ provides a higher modulus of rupture, in the range of 11.1 N mm^−2^ to 17.1 N mm^−2^. The rising average density is clearly displayed on the density profiles of the tested panels. The statistically significant differences in MOR mean values have been found for panels 13 and panels 11 and 15, as well as between panels 14 vs. 11 and 15. The comparison of MOR values of panels of different fibers type in face layers shows no statistically significant differences (panels 16 and 17). A slightly higher MOR of panel 17 (fiber type 2.); however, against the theory of fibers size contribution to panels’ mechanical properties [2] is connected to the high density of the near-surface layer. In general, the rising bending strength of the hardboards produced by the fibers of increasing freeness has been confirmed by [28]. The analysis of the influence of the density of panels made of fiber type 2 (panels 18–21) on MOR values shows a significant rise of MOR (from 10.9 N mm^−2^ to 19.5 N mm^−2^) with the density rise. The effect of higher densification of the fibrous zone of the panels has also been confirmed by [5]. Except for the highest value, the remaining MOR values of mentioned panels are comparable to MOR values of panels of similar densities, made of fiber type 1. In addition, the density profiles of mentioned panels proportionally rise. Statistically significant differences in mean MOR values were found for panels 18 vs. 20 and 21, as well as between panels 21 versus panels 18–20. When comparing the achieved results of MOR to requirements of EN 312 standard [27] for P2-type panels (interior furnishing purposes, including furniture), a minimum of 11 N mm^−2^, it can be said that it is hard to state whether single-layer panels (1–4) meet the mentioned requirements, due to the high irregularity of MOR values. The standard requirements have not been met by panels 5 (no fibers in face layers of the 3-layer panel), 13, and 18 (the lowest density).

The results of the modulus of elasticity investigation are presented in Figure 4. As can be seen, the relations between the tested factors influencing MOE are the same as in the case of the MOR investigation. There, the minimum MOE requirement, according to [27], is 1600 N mm^−2^.

### 3.2. Internal Bond and Density

The results of the internal bond investigation are displayed in Figure 5. The tests have shown that in the case of single-layer panels, the increase of fiber content to 50% and 75% (panels 3 and 4) caused a significant reduction of IB to the level of 0.13 N mm^−2^ (the lowest value within all 21 tested panels), where the IB of panel 1, without fibers addition was 0.46 N mm^−2^. There is no clear relation between changed fiber content and IB. However, a slight reduction of the core layers’ density was found, from 592 kg m^−3^ for panel 1 to 566 kg m^−3^ for panel 4 (Figure 6). The addition of fibers to 3-layer panels, with 16% face layers contribution, caused the increase of IB from 0.44 N mm^−2^ for panel 5 to 0.47 N mm^−2^ for panel 8. No statistically significant differences here for IB. Since the internal bond of the panels depends on their density in core layers, the density profiles should be analyzed. Regarding panels 5–8, the only significant changes in density profiles can be found in the face layers. The average density of core layers was 570 kg m^−3^. Since these panels were of 3 layers, and the fibers were added to face layers only, the fibers present in the face layers should not influence the IB. A slight rise (statistically insignificant) of IB with rising fibers share has been found for panels 9–12, where the face layers share was 32%. However, the mentioned IB average values, from 0.41 N mm^−2^ to 0.46 N mm^−2^, were slightly lower than in the case of the same fiber content but a lower contribution of face layers (panels 5–8). The core layers density of panels 9–12 rose from 552 to 579 kg m^−3^ (Figure 6). An obvious rise of IB was noted for panels 13, 14, 11, and 15, where the nominal density rose. The IB increase was from 0.26 N mm^−2^ for panel 13 (of the lowest density) to 0.42 N mm^−2^ for panel 15 of the highest density. The fiber content was 50% and was on a constant level here. The statistically significant differences in mean IB values have been found for panels 13 vs. 14 and 15. In addition, the density of core layers was raised for mentioned panels (Figure 6). No statistically significant differences in average values of IB were found between panels 16 (0.43 N mm^−2^) and 17 (0.45 N mm^−2^), where the different fibers were applied. A high difference in IB was noted when changing the density of panels made with the use of fibers type 2. As shown in Figure 5 for panels 18–21, the IB rises from 0.25 N mm^−2^ for the panel of the lowest density to 0.55 N mm^−2^ for the panel of the highest density. That IB change was more spectacular than for panels 13–15, where the fibers type 1 were applied. However, the density of the core layers of panels 13–15 and 18–21 were similar (Figure 6). These findings about the positive influence of the fibers’ presence on the internal bond under constant density align with [7]. There were no statistically significant differences between the IB of panels 19 and 20. According to [29], IB is strongly connected to the panels’ density and rises with density increase.

If referring the achieved IB values to minimal requirements for panels for interior equipment and furniture (0.35 N mm^−2^ for P2 type according to [27]), it should be said that for single-layer panels, the fiber content is over 25% causing IB reduction below standard requirements. Concerning the remaining panels, all the produced composites of density not lower than 620 kg m^−3^ meet the requirements of [27]. According to [30], a stronger influence on the internal bond can be found in glue content.

### 3.3. Surface Soundness

The results of the surface soundness measurement are displayed in Figure 7. It can be said that the relation of the factors investigated here in the case of SS is almost the same as in the case of IB. This is confirmed for single-layer panels (panels 1–4), where significantly lower values of SS were found for panels 3 and 4. For panels 5–8, where fibers were added to face layers, except panel 6, where the spread of individual results was large (high error bars—standard deviation), the average SS rises with fibers content rise. However, these are statistically insignificant changes in SS. The panels of 32% of face layers share to show the highest SS with 25% and 50% contribution of fibers in the face layers. These SS average values are statistically significantly different in panel 9. The further increase of fibers to 75% leads to a radical reduction of SS, even below the SS value for the reference panel (panel no. 9). Additionally, there is a statistically significant difference in SS between panels 10 and 12. The SS of panels made with the use of fibers type 1 remarkably rises when the density rises (panels 13, 14, 11, 15). Interestingly, there is a noticeable difference (however, statistically insignificant) in SS for panels made of different types of fibers. The SS of panel 16 with fiber type 1 is 0.7 N mm^−2^, whereas the SS of panel 17 with fiber type 2 is 1.2 N mm^−2^. This means over 71% higher value. In addition, the rise of the density of panels 18–21 caused the significant SS rise only between panels 18 and 19 when further density increase did not lead to higher SS.

The similarity of SS and IB results can be explained by the method of testing (load applying) of the samples. In both tests, the force works perpendicular to the wide surface of the panel, trying to pull the material. As far as in the case of IB, the sample breaks mostly in the middle of thickness, where the lowest density can be found; in the case of SS, the zone around the pulling equipment is cut off partially through the depth of the face layer of the highest density so that the break occurs still in the panel structure, but sometimes not exactly in the highest density zone. Considering that the density of panels was higher in the near-surface zone than in the middle of the thickness, as has been shown on density profiles, the SS values are significantly higher than IB. These findings, also connected to the density distribution over the panel thickness, have been confirmed by the previous research [30]. According to Sala et al. [31], the surface soundness of the high-density panels (HDF) can vary depending on the content of the fine particles in the face layers, which came from the addition of post-consumer (recycled) HDF panels.

Referring the achieved SS results to the requirements of [27] turns out that the following panels did not meet the standard (minimum 0.8 N mm^−2^ required): single layer panels of 50% and 75% fiber content (panels 3 and 4), panel 6, however, there is a high fluctuation of individual results of this mean value (error bars); 5-layer panel 16 of 50% of fiber type 1.

### 3.4. Surface Roughness

In Figure 8, the results of the surface roughness of the tested panels have been presented. According to these results, in the case of single-layer panels, the increase of fiber content to 50% and 75% (panel types 3 and 4) cause a significant reduction of both maximum roughnesses of peaks—Rpm and valleys—Rvm. When comparing panels 1 and 4, the reduction is over 60% for Rpm and over 82% for Rvm. Considering the Rvm decrease, it can be said that the surface is less porous, and this should lead to the reduction of paint and varnish demand reduction during surface finishing. Statistically significant differences have been found here between panels type 1 and 2 against 3 and 4. No clear relations have been found between the addition of fibers to 3-layer panels, with 16% face layers contribution (panel type 5–8). Statistically significant differences have been found between panels 5 and 7. A statistically significant decrease of Rpm and Rvm has been found for panels type 9–12, where rising of the fibers share were applied in the panels of 32% face layers share. The lowest values of Rpm and Rvm have been found for panel type 17, where the 5-layer panel with 25% of industrial SB fibers on the surfaces had been tested. The main factor influencing the lower values of surface roughness, which have been noted for panels with the surface structure modified by fibers type 2, is the fact that these fibers were of fine size. This was mentioned in the Materials and Methods section that the fibers type 2. have a higher defibration degree—63 DS, referred to as 47 DS for fibers type 1. The influence of the size of fibers has been confirmed by [31], where the various share of spruce fibers have been added to HDF production. According to [4], the surface roughness of particleboards, in general, can be radically improved by the addition of fibrous materials. In addition, it should be mentioned that the best surface roughness properties source can be found in the density profile (Figure 9e), where the high-density zone is located near the surface.

### 3.5. Density Profiles

The representative density profiles of the tested panel have been presented in Figure 9. To improve the plots’ readability, the density profiles have been presented to the middle of the panel thickness since these are symmetrical. Figure 9a presents the profiles of the panel types 1–4, where the fibers were added to the entire structure of single-layer panels. The unexpected rise of the face zone density has been found for panel type 2, where the fiber content was 25%. A similar, high density in the same zone has been found for panel type 4, with 75% fiber content in the entire structure. When analyzing the distribution of the density over the panel thickness for panels type 5–8 (3-layer, 16% of face layers share with increasing fibers content in face layers) (Figure 9b), it should be said that the increase of fibers shares is clearly visible in rising densification of face layers. The peak density can be found in 0.8–1 mm of the panel thickness. The significant rise of the face layer density with increasing fibers’ content has also been found for panels type 9–12 (Figure 9c), where the face layers share was 32%. Since the contribution of the face layers was higher than in the case of the panels’ type 5–8, the peak density has shifted to 1–1.5 mm of the panel thickness. When analyzing the influence of the panel density on the distribution of the density over the panel thickness (Figure 9d) for panels types 13, 14, 11, and 15, where the density increased from 580 kg m^−3^ to 710 kg m^−3^, and where the fibers’ content was 50% in the face layers, it can be concluded that the density profiles are shifted proportionally to the panels’ density rise, with the higher shift of panel type 15, where the density rise was 50 kg m^−3^ instead of 40 kg m^−3^ as it was for panels type 14 and 11. The clear difference between the density profiles of the panels with the content of different fibers type (1 and 2) has been found in Figure 9e. As it has been found, the fibers type 2, due to the higher defibration degree, resulted in the higher density of the face layers closer to the panel 17 surface. This densification can contribute to the high modulus of rupture and modulus of elasticity and can act as an intermediate layer [3]. The distribution of the density of panels 18–21 (Figure 9f), where the panels’ density has changed, was similar to these of panels in Figure 9d. However, the slightly higher density of panel 21 face layers has been found. This has been caused by the presence of fine (of higher defibration degree) fibers in the face layers.

## 4. Conclusions

The conducted tests showed that single-layer particleboards made of wood fibers and particles do not meet the standard requirements specified in EN 312 [27] for P2-type boards (intended for interior furnishing, including furniture). The properties of the boards are very diverse. Moreover, it is difficult to define the factors determining their level. The main reason for the unsatisfactory mechanical quality of the boards can probably be the uneven resination of particles and fibers. Applying the glue to the mixture of particles of such different sizes caused agglomeration of the fibers and, thus, a heterogeneous structure of the panels.

The above phenomenon also occurred, but to a much lesser extent, during the blending of the mixture of fine particles and fibers used for the face layers of the three-layer boards.

The properties of the three-layer boards are closely related to the density of the boards, as well as to the face layers and fiber share. The influence of the share of layers on static bending and the modulus of elasticity is also clear. It is visible mainly in the case of lower board density, as well as the influence of the type of fibers used. The application of the fibers intended for MDF production allowed the achievement of higher mechanical parameters of the panels due to the higher densification of the face layers. Additionally, better surface roughness characteristics have been found for panels with these fibers. The test results showed that the production of three-layer boards meeting the requirements of the above standard should not pose any difficulties; it is even possible to obtain boards with much higher strength than required or with standard properties, with a density lower than the average for the currently produced furniture boards. That shows the potential of particleboard density reduction by fiber incorporation without losing the crucial features of the panels.

The five-layer boards meet the standard requirements to the full extent, but the continuation of research on their use does not seem justified. The strength of the near-surface layer (surface soundness) is lower than that obtained for three-layer panels, and the manufacturing process is more complicated and, thus, less effective.

The influence of the share of the fibers type 1 (lower degree of defibration) on properties of the single layer panels of constant density. There is a significant improvement in the mechanical properties with a 25% fiber contribution. This is because of the optimal filling of the empty zones between the large-size particles of single-layer panels. However, the surface roughness (Rpm) is the worst for these panels. The further increase of fibers’ content causes improvement of surface roughness. In the case of the 3-layer panels, where the contribution of face layers in the entire panel is 16%, the rising amount of fibers in face layers to 75% causes over 50% rise of MOR. There is also an increase in MOE and SS, as well as an improvement in surface roughness. The improvement of IB was not so spectacular. In the case of the 3-layer panels, where the content of the outer layers was 32%, the addition of fibers to the face layers from 0% to 75% led to improvement of MOR and MOE, as well as surface roughness; however, the best bending and SS parameters have been found for 50% fibers’ addition, and 75% for best surface roughness. Changing the density of the 3-layer panels of constant 50% content of fibers of a lower degree of defibration in face layers, from 580 kg m^−3^ to 710 kg m^−3^, radically improves all the mechanical features of the panels, with the minimal improvement of surface roughness. The same test, but with the use of fibers of a higher degree of defibration, leads to an improvement of all the tested mechanical properties, as well as surface roughness, even if referred to panels with a lower degree of defibration. When comparing the influence of the type of the fibers used with the remaining constant panels’ production parameters, as 100% fibrous face layers in 5-layer panels, it can be said that there was no significant influence of fibers type on panels’ mechanical parameters, like MOR, MOE, and IB, but the important rise of SS and surface roughness improvement has been noted. The better surface roughness is due to the fact that the fibers of a higher degree of defibration (finer) have been used, as well there was higher densification of face layers found on the density profile. What is important here, the fibers of a lower degree of defibration, which are of higher strength and thus intended for medium-density fiberboard production, did not improve the MOR and MOE properties. It can be concluded that in the case of single-layer panels, the fiber content is an important factor influencing the panels’ performances. In the case of 3-layer panels, there is an optimal 50% ratio of fiber content. It is preferable to use fibers of a higher degree of defibration when producing fibrous-particle panels.

The novelty of the conducted research is the attempt to modify the structure of particleboards by adding various amounts of two different types of fibers by mixing them with particles or adding them as separate layers and producing panels of different densities.

## Figures and Tables

**Figure 1 materials-15-08530-f001:**
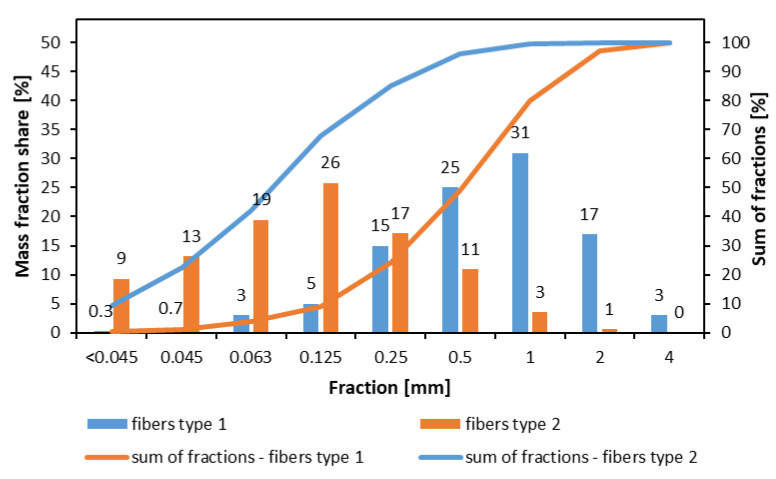
The morphological characteristics of the fibers.

**Figure 2 materials-15-08530-f002:**
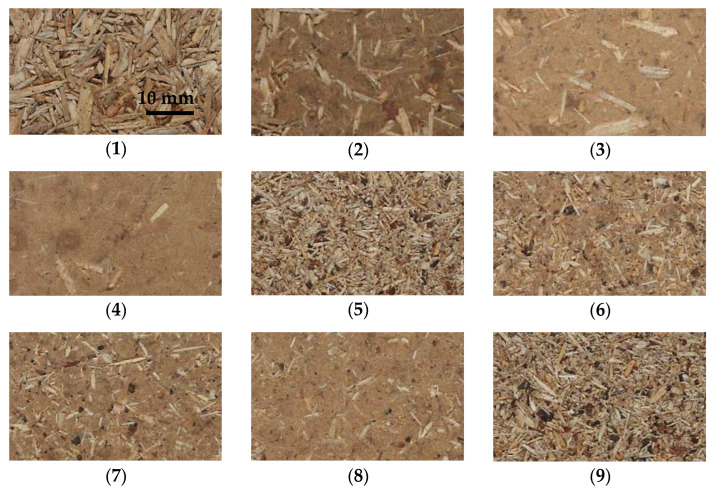
The pictures of the morphology of tested composite surface (the numbers in brackets refer to panel type—see Table 1; pictures dimensions: 30 mm (vertical) × 50 mm (horizontal)).

**Figure 3 materials-15-08530-f003:**
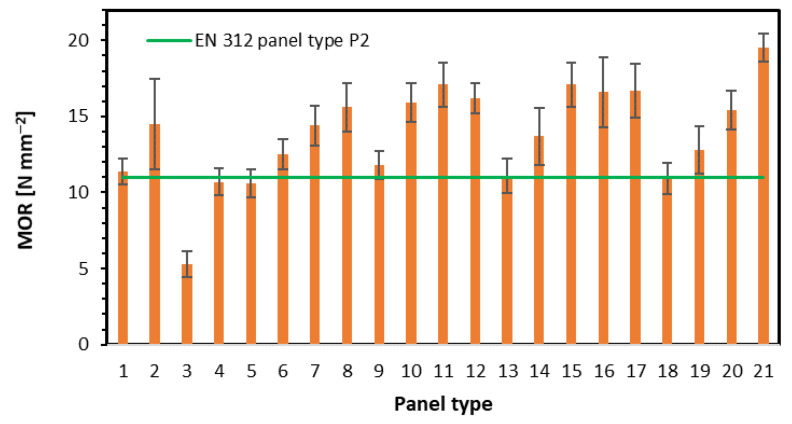
The modulus of rupture of tested panels.

**Figure 4 materials-15-08530-f004:**
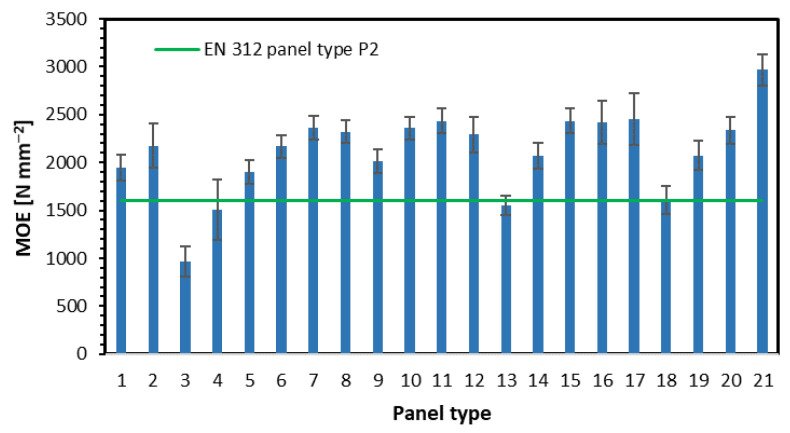
The modulus of elasticity of tested panels.

**Figure 5 materials-15-08530-f005:**
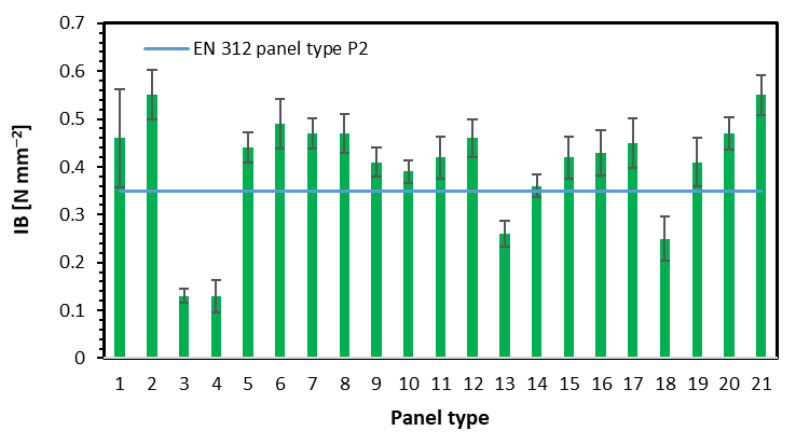
The internal bond of tested panels.

**Figure 6 materials-15-08530-f006:**
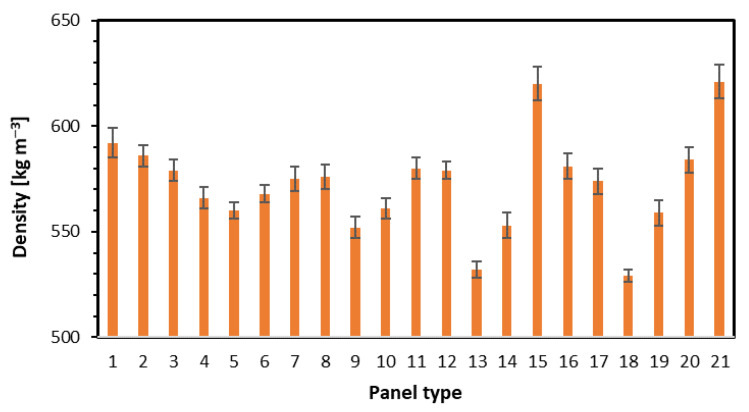
The density of the core layers (middle of thickness) of tested panels.

**Figure 7 materials-15-08530-f007:**
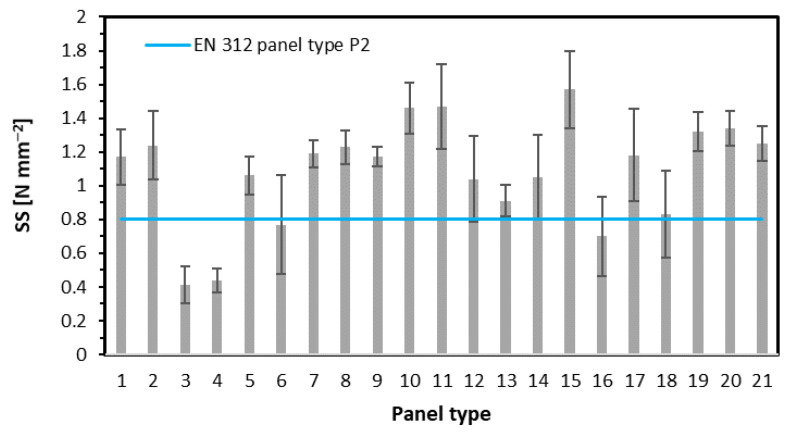
The surface soundness of tested panels.

**Figure 8 materials-15-08530-f008:**
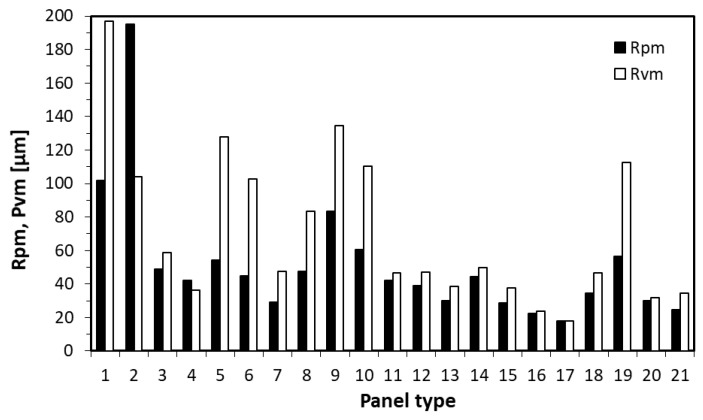
The values of peaks and valley surface roughness of tested composites.

**Figure 9 materials-15-08530-f009:**
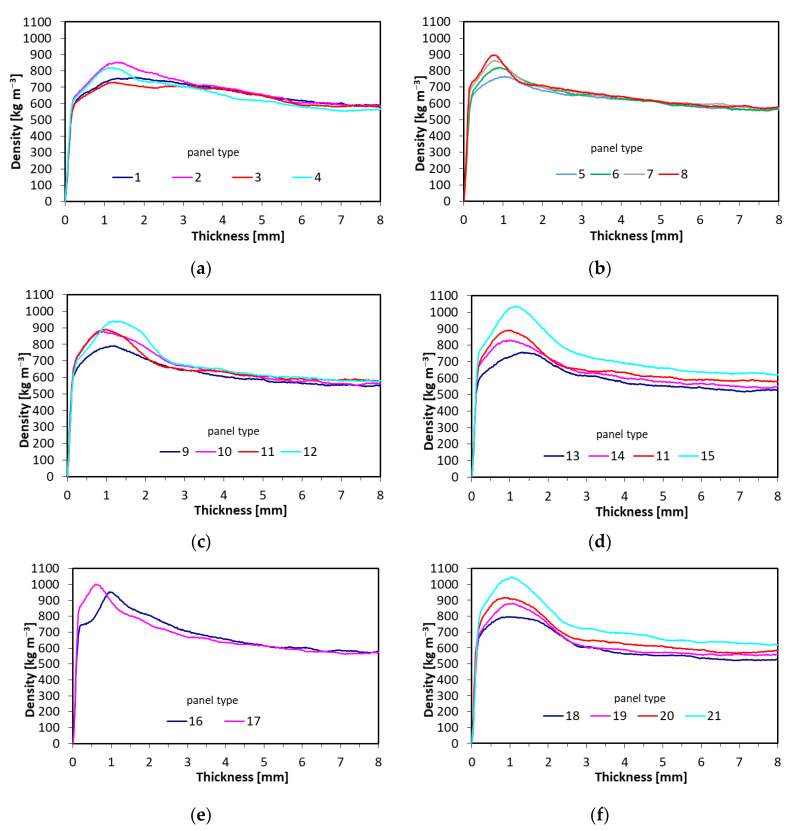
Density profiles of tested panels: (**a**) single layer, various share of fibers, (**b**) 3-layer panels, 16% face layers’ share, (**c**) 3-layer panels, 32% face layers’ share, (**d**) 50% fibers’ type 1. share, various densities, (**e**) different type of fibers when blended with glue separately, fibers and particles, (**f**) 50% fibers’ type 2. share, various density.

**Table 1 materials-15-08530-t001:** Types of tested panels.

Panel Type	Number of Layers	Nominal Density [kg m^−3^]	Face Layers’ Share [%]	Fiber Type **	Fibers’ Share in Face Layers [%]
1	1	660	-	1	0
2					25
3					50
4					75
5	3		16		0
6					25
7					50
8					75
9			32		0
10					25
11					50
12					75
13		580			50
14		620			50
15		710			50
16 *	5	660			25
17 *				2	25
18	3	580			50
19		620			50
20		660			50
21		710			50

* fibers and particles blended with glue separately; added as a separate outer fiber layer ** 1—industrial MDF fibers; 2—industrial SB fibers.

**Table 2 materials-15-08530-t002:** The average values of achieved results of tested panels.

Panel Type	MOR	MOE	IB	SS	Density **	Rpm	Rvm
	[N mm^−2^]				[kg m^−3^]	[µm]	
1	11.4 (0.86) *	1943 (136.5)	0.46 (0.10)	1.17 (0.16)	592 (7)	102.0	197.0
2	14.5 (3.00)	2173 (231.3)	0.55 (0.05)	1.24 (0.20)	586 (5)	195.0	104.0
3	5.3 (0.83)	967 (153.7)	0.13 (0.02)	0.41 (0.11)	579 (5)	49.0	58.7
4	10.7 (0.90)	1503 (314.6)	0.13 (0.03)	0.44 (0.07)	566 (5)	42.1	36.3
5	10.6 (0.94)	1901 (127.1)	0.44 (0.03)	1.06 (0.11)	560 (4)	54.4	127.6
6	12.5 (0.99)	2166 (122.4)	0.49 (0.05)	0.77 (0.29)	568 (4)	44.8	102.6
7	14.4 (1.29)	2363 (119.0)	0.47 (0.03)	1.19 (0.08)	575 (6)	29.3	47.5
8	15.6 (1.61)	2320 (118.3)	0.47 (0.04)	1.23 (0.10)	576 (6)	47.5	83.6
9	11.8 (0.95)	2017 (123.4)	0.41 (0.03)	1.17 (0.06)	552 (5)	83.6	134.5
10	15.9 (1.27)	2359 (115.1)	0.39 (0.02)	1.46 (0.15)	561 (5)	60.4	110.5
11	17.1 (1.46)	2435 (131.8)	0.42 (0.04)	1.47 (0.25)	580 (5)	42.3	46.6
12	16.2 (0.97)	2290 (183.7)	0.46 (0.04)	1.04 (0.25)	579 (4)	39.1	46.9
13	11.1 (1.13)	1550 (104.9)	0.26 (0.03)	0.91 (0.09)	532 (4)	30.2	38.6
14	13.7 (1.87)	2071 (137.7)	0.36 (0.02)	1.05 (0.25)	553 (6)	44.6	49.6
15	17.1 (1.46)	2435 (131.8)	0.42 (0.04)	1.57 (0.23)	620 (8)	28.6	37.8
16	16.6 (2.30)	2418 (221.5)	0.43 (0.05)	0.70 (0.24)	581 (6)	22.6	23.7
17	16.7 (1.75)	2457 (268.7)	0.45 (0.05)	1.18 (0.27)	574 (6)	18.0	18.1
18	10.9 (1.04)	1611 (145.9)	0.25 (0.05)	0.83 (0.26)	529 (3)	34.7	46.6
19	12.8 (1.54)	2074 (154.7)	0.41 (0.05)	1.32 (0.11)	559 (6)	56.4	112.4
20	15.4 (1.28)	2337 (139.3)	0.47 (0.03)	1.34 (0.10)	584 (6)	30.0	31.6
21	19.5 (0.92)	2966 (162.9)	0.55 (0.04)	1.25 (0.11)	621 (8)	24.6	34.7

* data in brackets are the standard deviation values; ** minimal density in the core layer.

## Data Availability

The data presented in this study are available on request from the corresponding author.

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
