# Peer review of "Selected Properties of Single and Multi-Layered Particleboards with the Structure Modified by Fibers Implication"

_materials, 2022, doi:10.3390/ma15238530_

Round 1

Reviewer 1 Report

Review report for manuscript "Particleboard Structure Modification by Fibers Implication". The manuscript is well written, only some minor corrections are needed before publication.

Line 37: Please explain Ra and Rz.

Line 39-40: Please add a reference to this statement.

In the Introduction, the authors should more deeply discuss the use of fibers in particleboard production in the past, which also influences the particleboard's physical and mechanical properties. There are plenty of sources published, for example:

doi.org/10.1016/j.indcrop.2015.09.079

doi.org/10.1016/j.matdes.2013.09.066

doi.org/10.1016/j.conbuildmat.2019.02.024

and many others.

Line 75-77: Please unify the aim of the manuscript here and in the abstract, as it is not exactly the same.

Lines 88-90: Please add more information about the properties of resin (UF), hardener content, press closing time, and press pressure.

Line 130: here modulus of rupture is mentioned, in the Abstract you mentioned bending strength, please unify.

The Results and discussion part are well written. I suggest adding the Novelty of your research (also to the abstract section).

Author Response

Teh response has been attached

Reviewer 3 Report

Dear authors!

The manuscript is well written. The topic is up to date and of high interest to potential readers, due to the research leading to the improvement of wood-based material surface quality and exploration of better and more optimal utilization of wood raw materials. The results are properly evaluated and presented. The discussion is broad and contains current literature citations.

Detailed remarks:

1. Add some number values to the Abstract
2. In the methodology section please add information on whether you were calibrating (by sanding) your panels after pressing or not.
3. Please correct the reference to the pictures of the panels’ surface given in line 102 (should be Figure 1).

Reviewer 4 Report

The mechanical properties of single or multi-layer particleboards modified by incorporation of wood fibers were studied in this manuscript. It is helpful for the production of particleboards. There are some problems for the work. My suggestions are as below.

1.     Why do the authors choose two types of fibers for the modification? What’s the difference between them? Please give us some introductions on the two types of fibers, for examples, the species, the appearance, the dimensions, and so on.

2.     The manuscript is badly organized. It is difficult to understand what the authors want to indicate, especially for the figures. The expression is not clear for reading and comparation. Please adjust the expression ways of the figures and make the testing panels into some groups.

Round 2

Reviewer 2 Report

Please correct the subtitle.

Good luck

Author Response

The manuscript title has been changed to:

Selected Properties of Single and Multi-layered Particleboards with the Structure Modified by Fibers Implication

Reviewer 4 Report

The manuscript actually isn’t improved a lot. My suggestions are as below.

1.     The authors said that the novelty of the conducted research is the attempt to use two types of fibers to modify the structures of single, three, and five particleboards. I don’t think it is a novelty. What’s the deeper reason or the explanation in theory for the possible difference between the boards modified with different types of fibers? What’s the objectives of the research? What is the important factor to affect the performances of the boards? Please say more in the introduction and discussion parts on the reasons for the difference among the particleboards. The conclusion parts have to answer the questions what effects of the different types of fibers on the final performances of the particleboards.

2.     Please summarize the boards’ performances results in one table for easily reading.
